# Epidemiology of hypertension in Northern Tanzania: a community-based mixed-methods study

Sophie W Galson,[1,2] Catherine A Staton,[1,2,3] Francis Karia,[4] Kajiru Kilonzo,[5] Joseph Lunyera,[6] Uptal D Patel,[7] Julian T Hertz,[1,2] John W Stanifer[2,8]

For numbered affiliations see end of article.

**Correspondence to**
Dr Sophie W Galson;
sophie.galson@duke.edu

## ABSTRACT

**Introduction** Sub-Saharan Africa is particularly vulnerable to the growing global burden of hypertension, but epidemiological studies are limited and barriers to optimal management are poorly understood. Therefore, we undertook a community-based mixed-methods study in Tanzania to investigate the epidemiology of hypertension and barriers to care.

**Methods** In Northern Tanzania, between December 2013 and June 2015, we conducted a mixed-methods study, including a cross-sectional household epidemiological survey and qualitative sessions of focus groups and in-depth interviews. For the survey, we assessed for hypertension, defined as a single blood pressure ≥160/100 mm Hg, a two-time average of ≥140/90 mm Hg or current use of antihypertensive medications. To investigate relationships with potential risk factors, we used adjusted generalised linear models. Uncontrolled hypertension was defined as a two-time average measurement of ≥160/100 mm Hg irrespective of treatment status. Hypertension awareness was defined as a self-reported disease history in a participant with confirmed hypertension. To explore barriers to care, we identified emerging themes using an inductive approach within the framework method.

**Results** We enrolled 481 adults (median age 45 years) from 346 households, including 123 men (25.6%) and 358 women (74.4%). Overall, the prevalence of hypertension was 28.0% (95% CI 19.4% to 38.7%), which was independently associated with age >60 years (prevalence risk ratio (PRR) 4.68; 95% CI 2.25 to 9.74) and alcohol use (PRR 1.72; 95% CI 1.15 to 2.58). Traditional medicine use was inversely associated with hypertension (PRR 0.37; 95% CI 0.26 to 0.54). Nearly half (48.3%) of the participants were aware of their disease, but almost all (95.3%) had uncontrolled hypertension. In the qualitative sessions, we identified barriers to optimal care, including poor point-of-care communication, poor understanding of hypertension and structural barriers such as long wait times and undertrained providers.

**Conclusions** In Northern Tanzania, the burden of hypertensive disease is substantial, and optimal hypertension control is rare. Transdisciplinary strategies sensitive to local practices should be explored to facilitate early diagnosis and sustained care delivery.

## Strengths and limitations of this study

► Rigorous study design based on random, community-based sampling.
► The mixed-methods approach allowed for triangulation from multiple data sources with reproducible methods.
► Barriers to optimal hypertension care were explored using qualitative studies with key informants from both biomedicine and traditional medicine practices.
► As this was a cross-sectional study, causal inferences cannot be drawn.
► Our role as biomedical practitioners may have limited our ability to interpret results (researcher bias) about differences in disease understanding, and our inferences are derived from a biomedical perspective.

## INTRODUCTION

Non-communicable diseases (NCDs), including hypertension, are a global epidemic disproportionately affecting health outcomes in low-income and middle-income countries (LMICs).[1–3] In sub-Saharan Africa (SSA) alone, more than 125 million people are expected to have hypertension by the year 2025.[4 5] Hypertensive-related complications are currently one of the leading causes of morbidity in SSA, and by 2030, hypertension and other NCDs are projected to surpass communicable diseases as the top cause of mortality.[2 3 5] Despite the overwhelming burden, SSA is mostly unprepared to address this impending public health crisis.[6 7]

Previously identified barriers to addressing hypertension in SSA include under-recognition, undertreatment and a limited understanding of its epidemiology.[8 9] In particular, the lack of reliable health statistics and a paucity of community-based epidemiological data limit the ability for detection, surveillance and creation of public health strategies for prevention and treatment.[7 10] In Northern Tanzania, for example, only 10%–20% of patients with previously detected hypertension

are receiving treatment, and only 16% of those on treatment were adequately controlled.[11 12]

Differences in hypertension care are related to several factors beyond healthcare access alone, including limited health literacy, cultural and social barriers and heuristically different health belief models.[13] As such, understanding the epidemiology of hypertension as well as the social and community barriers to optimal care is critical for developing prevention and treatment strategies; however, few such studies have been conducted. Therefore, as part of the Comprehensive Kidney Disease Assessment for Risk Factors, Epidemiology, Knowledge and Attitudes (CKD-AFRiKA) study, we conducted a mixed-methods study in order to characterise the community-based epidemiology of hypertension and barriers to optimal care through exploration of patient-centred and community-centred perspectives.[14 15]

## METHODS
### Ethics statement
The study protocol was approved by the Duke University Institutional Review Board (no Pro00040784), the Kilimanjaro Christian Medical College Ethics Committee (EC no 502) and the National Institute for Medical Research in Tanzania. Written informed consent (by signature or thumbprint) was obtained from all participants, and all participants with abnormal findings received counselling, educational pamphlets and reimbursement with referral for follow-up. All participants were reimbursed between TZS1500 and TZS12 000 (approximately US$0.75–US$5.00) depending on their distance of travel.

### Study setting
We conducted a mixed-methods study between December 2013 and June 2015 in the Kilimanjaro Region of Tanzania. The adult regional population is greater than 900 000 people, and it has a female majority (58%).[16] Almost 35% of the adult population lives in an urban setting, which is comparable to national estimates, and the HIV prevalence is 3%–5%.[16 17] The unemployment rate is 19%, and most people have only a primary education (77%).[18] The median age, average household size and occupation distribution are similar to national estimates.[16–18] The largest ethnic group is the Chagga tribe, and Swahili is the major language.[16] The region comprises seven districts; our study was conducted in the Moshi Urban and Moshi Rural districts, which were selected based on their representative populations and proximity to our research infrastructure.[16]

### Quantitative sampling and data collection
Detailed sampling methods for the CKD-AFRiKA study have been previously reported.[14 15 19] In brief, a three-stage cluster probability sampling method, stratified by urban and rural settings, was used to randomly select neighbourhoods based on probability proportional to size. Within each selected neighbourhood, a cluster site was determined using geographic points randomly generated by Arc Global Information Systems, V.10.2.2 (Environmental Systems Research Institute, Redlands, California, USA). From the cluster site, households were then randomly chosen based on both a coin-flip and die-rolling technique according to our established protocol.[14]

All non-pregnant, community-dwelling adults (age ≥18 years) from the selected households were recruited into the study. The sample size was designed to estimate the prevalence of chronic kidney disease with a precision of 5% when accounting for the cluster-design effect. To reduce non-response rates, a minimum of two additional visits were attempted during off-hours (evenings and weekends) as well as multiple phone calls using mobile phone numbers.

All data were collected using trained, local surveyors. Each participant completed a demographic and medical history survey, which included self-reported history of diabetes, hypertension, HIV, kidney disease and heart disease (coronary, structural or heart failure). If participants were receiving biomedical treatment in the form of medical therapy, specific drug information was collected. Women additionally gave a self-reported history for pregnancy or menstruation. Awareness was defined as giving a self-reported history of hypertension and subsequently testing positive for hypertension in our screening process.

Anthropometric data (including height, weight and body mass index (BMI)) were recorded for each participant. Normal weight was defined as a BMI of 20 to $24.9 \, kg/m^2$. Overweight was defined as a BMI $\geq 25 \, kg/m^2$, and obesity was defined as a BMI $\geq 30 \, kg/m^2$. We measured blood pressure using the automated Omron HEM-712 sphygmomanometer (Omron Healthcare, Bannockburn, Illinois, USA) that has an adjustable cuff size. The machine was calibrated monthly during data collection. All participants were seated in an erect position with feet flat on the floor for a minimum of 5 min before measurements. Two measurements separated by >5 min were then performed. Hypertension was defined as a single blood pressure measurement of greater than 160/100 mm Hg, a two-time average measurement of greater than 140/90 mm Hg or current self-reported use of antihypertensive medications. Uncontrolled hypertension was defined as a two-time average measurement of greater than 160/100 mm Hg irrespective of treatment status. Tobacco use and alcohol use were defined as self-reported current ongoing use, former use or never used.

### Qualitative data collection
To explore patient-centred and community-centred perspectives related to barriers in optimal hypertension care, we conducted focus group discussions (FGDs) and in-depth interviews in a central, easily accessible location. These sessions have been described previously.[20] In brief, we conducted FGDs and in-depth interviews with key informants from the community including well adults from the

general population, chronically ill adults receiving care at the hospital medicine clinics, adults receiving care from traditional healers and adults purchasing traditional medicines from herbal vendors, traditional healers and medical doctors. Purposive sampling was used to recruit the key informants. We targeted men and women of all ages from urban and rural settings with different education levels and ethnicities. FGDs were held in a rented office space in Moshi Urban that was well-known and easily accessible to local residents and ensured privacy. Each FGD lasted between 4 and 6 hours including breaks. In-depth interviews were conducted at the same office space with the exception of the traditional healers and herbal vendors who were interviewed at their places of work; these sessions lasted 1 to 2 hours. All sessions were semi-structured, open-ended and probing. The discussion guide was initially written in English and then translated to Swahili by an independent team. All sessions were moderated by a native, local member of our team (FK). All sessions were audio-recorded, and two note-takers transcribed and independently translated each session. A moderator then reviewed the transcripts to ensure accuracy. Debriefings were held after each session, and team meetings were again held following translation.

### Quantitative data analysis

The mean and SD or median and IQR were reported for continuous variables. Prevalence estimates were sample-balanced using age and gender weights based on the 2012 urban and rural district-level census data. We used a $X^2$ test or Fisher's exact test to compare differences between groups. All p values are two-sided at a 0.05 significance level. Quantitative data were analysed using STATA V.14 (STATA).

A secondary aim of the analysis was to explore associations between hypertension and potential risk factors related to lifestyle. Crude and adjusted prevalence risk ratios (PRRs) were estimated using generalised linear models with a log link, and we used Taylor Series linearisation to account for the design effect on variance due to cluster sampling. Separate univariable and multivariable models were fitted to hypertension status for each lifestyle-related variable including alcohol use, tobacco use, traditional medicine use, living in an urban environment and overweight/obesity status. Models were adjusted for confounding factors potentially associated with hypertension and each potential risk factor, including age, gender and ethnicity. We did not include education or occupation in our models due to a priori assumptions about their potential upstream causal association with lifestyle-related risk factors.

All quantitative data were collected on paper and then electronically entered into and managed using REDCap electronic data capture tools hosted at Duke University.[21] All data were verified after electronic data entry by an independent reviewer to ensure accuracy.

### Qualitative data analysis

We conducted a thematic analysis of the qualitative data by applying an inductive approach to the framework method.[22] The approach was based, in part, on our previously developed model which explored determinants of traditional medicine use and biomedical healthcare utilisation among individuals in Kilimanjaro with NCDs, including hypertension.[20] After data reduction, we performed open-coding of all transcripts. We used a 'cultural insider' (emic) and a 'cultural outsider' (etic) to independently code the data. The cultural insider was a native researcher living in the region (FK), and the cultural outsider was a researcher foreign to the region (JS). Comparisons were made between each code set, and areas of disagreement were discussed and resolved by revisiting the data. This approach allowed us to explore concepts that otherwise may have been overlooked or misinterpreted by either researcher individually. The qualitative coding, analytic memos and corresponding matrices were stored and analysed using NViVO V.10.0 (QRS International, Melbourne, Australia). The codes were grouped together into categories, and we used a coding index to formulate connections and explore relationships.

## RESULTS
### Study populations

We enrolled 481 adults into the quantitative study (table 1). The median age was 45.0 years (IQR 35–59). The majority of participants were women (n=358; 74%), lived in an urban location (n=370; 77%), ethnically Chagga (n=288; 60%) and only had a primary school education (n=349; 73%). The most common occupation among participants was farming or daily wage work (n=199; 41%). Many participants reported ongoing use of alcohol (n=198; 41%) and traditional medicine use over the previous year (n=272; 57%), with the most commonly reported frequency of traditional medicine use at 1–5 times (31.0%) per year. Among participants currently using prescribed biomedicines (n=70), the proportion of participants reporting traditional medicine use was more substantial, with 69% (n=48) reporting traditional medicine use over the previous year.

The household non-response rate was 15.0%, and the individual non-response rate was 20.6%. Compared with the regional population,[16] men (P<0.001) and young adults 18–39 years (P=0.001) were more likely to be non-responders in our study, and the proportion of participants with a secondary or postsecondary education (22%) was higher than the regional average (15%) (P=0.02). We observed no significant differences in occupation between the responders and non-responders (P=0.64).

In the qualitative study, we conducted 5 FGDs and 11 in-depth interviews (table 2). FGDs and in-depth interviews included even numbers of men (n=35; 50%)

**Table 1** Baseline characteristics for the quantitative study

| Variable (n, %) | Total (n=481) | Normotensive (n=332) | Hypertensive (n=149) | P value |
|---|---|---|---|---|
| Gender | | | | 0.12 |
| Male | 123 (25.6%) | 78 (23.5%) | 45 (30.2%) | |
| Female | 358 (74.4%) | 254 (76.5%) | 104 (69.8%) | |
| Age | | | | <0.01 |
| 18–39 years | 172 (35.8%) | 152 (45.8%) | 20 (13.4%) | |
| 40–59 years | 191 (39.7%) | 132 (39.8%) | 59 (39.6%) | |
| 60+ years | 118 (24.5%) | 48 (14.5%) | 70 (47.0%) | |
| Ethnicity | | | | 0.40 |
| Chagga | 288 (59.9%) | 193 (58.1%) | 95 (63.7%) | |
| Pare | 66 (13.7%) | 51 (15.4%) | 15 (10.1%) | |
| Sambaa | 27 (5.6%) | 20 (6.0%) | 7 (4.7%) | |
| Other* | 100 (20.8%) | 68 (20.5%) | 32 (21.5%) | |
| Education | | | | <0.01 |
| None | 31 (6.4%) | 11 (3.31%) | 20 (13.4%) | |
| Primary | 349 (72.6%) | 246 (74.1%) | 103 (69.1%) | |
| Secondary | 74 (15.4%) | 54 (16.3%) | 20 (13.4%) | |
| Postsecondary | 27 (5.6%) | 21 (6.3%) | 6 (4.03%) | |
| Occupation | | | | <0.01 |
| Unemployed† | 74 (15.4%) | 55 (16.6%) | 19 (12.8%) | |
| Farmer/wage earner | 199 (41.4%) | 135 (40.7%) | 64 (43.0%) | |
| Small business/vendors | 158 (32.8%) | 121 (36.5%) | 37 (24.8%) | |
| Professional‡ | 50 (10.4%) | 21 (6.3%) | 29 (19.5%) | |
| Lifestyle practices | | | | |
| Ongoing tobacco use | 50 (10.4%) | 34 (10.2%) | 16 (10.7%) | 0.87 |
| Ongoing alcohol use | 198 (41.2%) | 121 (36.4%) | 77 (51.7%) | 0.02 |
| Traditional medicine use | 272 (56.6%) | 196 (59.0%) | 76 (51.0%) | 0.10 |
| Self-reported medical history | | | | |
| Diabetes | 61 (12.7%) | 29 (8.7%) | 32 (21.5%) | <0.01 |
| Hypertension | 134 (28.0%) | 62 (18.8%) | 72 (48.3%) | <0.01 |
| Stroke | 8 (1.7%) | 2 (0.6%) | 6 (4.0%) | 0.01 |
| Heart disease§ | 18 (3.7%) | 7 (2.1%) | 7 (4.7%) | 0.08 |
| Kidney disease | 14 (2.9%) | 10 (3.0%) | 4 (2.7%) | 0.84 |

*Other tribal ethnicities represented in our groups include Luguru, Kilindi, Kurya, Mziguwa, Mnyisanzu, Rangi, Jita, Nyambo, and Kaguru
†Includes housewives and students
‡Professional includes any salaried position (eg, nurse, teacher, government employee, etc) and retired persons
§Heart Disease includes coronary disease, heart failure, or structural diseases

and women (n=35; 50%) and had an age range of 18 to 74 years. Most participants were of the Chagga ethnic group (n=37; 53%) and were Roman Catholic (n=29; 41%), but Islamic (n=11; 16%), Lutheran (n=17; 24%) and Christian Evangelical (n=11; 16%) were also represented as well as 13 different tribal ethnicities. Education levels varied from none (n=2; 3%) to university level (n=13; 19%), but the majority had only completed a primary education (n=38; 54%). Most participants were from urban residences (n=55; 79%).

### Burden of hypertension

The prevalence of hypertension was 28% (95% CI 19.4% to 38.7%). The design effect of the cluster sampling was 2.34, with a neighbourhood-level Intra-cluster Correlation Coefficient (ICC) coefficient of 0.075. Mean systolic blood pressure was 129.5 mm Hg (SD 24.3), mean diastolic blood pressure was 77.6 mm Hg (SD 12.2) and the median age was 58 years (IQR 45–65). The prevalence of hypertension was 59.3% (95% CI 50.1% to 67.9%) in participants ≥60

**Table 2** Baseline characteristics for the qualitative study

| Study population | FGD1<br>Clinic patients | FGD2<br>General population | FGD3<br>Clinic patients | FGD4<br>General population | FGD5<br>Medical doctors | In-depth interviews<br>Patients from healers and vendors |
|---|---|---|---|---|---|---|
| Participants (N) | 15 | 12 | 16 | 12 | 4 | 11 |
| Gender | | | | | | |
| Male | 0 (0%) | 0 (0%) | 16 (100%) | 12 (100%) | 2 (50%) | 5 (45%) |
| Female | 15 (100%) | 12 (100%) | 0 (0%) | 0 (0%) | 2 (50%) | 6 (55%) |
| Age range (years) | 25–61 | 26–65 | 18–70 | 18–74 | 30–36 | 19–60 |
| Ethnicity | | | | | | |
| Chagga | 11 (73%) | 9 (75%) | 11 (69%) | 4 (33%) | 2 (50%) | 2 (18%) |
| Pare | 2 (13%) | 2 (17%) | 2 (13%) | 5 (42%) | 0 | 0 |
| Maasai | 0 | 0 | 0 | 0 | 0 | 4 (36%) |
| Sambaa | 1 (7%) | 1 (8%) | 1 (6%) | 0 | 0 | 3 (27%) |
| Other* | 1 (7%) | 0 | 2 (13%) | 3 (25%) | 2 (50%) | 2 (18%) |
| Education | | | | | | |
| None | 0 | 0 | 0 | 0 | 0 | 2 (18%) |
| Primary | 11 (73%) | 10 (83%) | 10 (63%) | 3 (25%) | 0 | 4 (36%) |
| Secondary | 3 (20%) | 2 (17%) | 5 (31%) | 6 (50%) | 0 | 1 (9%) |
| University | 1 (7%) | 0 | 1 (6%) | 3 (25%) | 4 (100%) | 4 (36%) |
| Occupation | | | | | | |
| Unemployed† | 2 (13%) | 4 (33%) | 0 | 1 (8%) | 0 | 3 (27%) |
| Student | 0 | 0 | 4 (25%) | 5 (42%) | 0 | 0 |
| Farmer/wage earner | 4 (27%) | 3 (25%) | 8 (50%) | 3 (25%) | 0 | 5 (45%) |
| Small business | 3 (20%) | 2 (17%) | 3 (19%) | 2 (17%) | 0 | 1 (9%) |
| Professional‡ | 4 (27%) | 3 (25%) | 1 (6%) | 1 (8%) | 4 (100%) | 2 (18%) |
| Religion | | | | | | |
| Roman Catholic | 5 (33%) | 5 (42%) | 8 (50%) | 1 (8%) | 3 (75%) | 7 (64%) |
| Lutheran | 6 (40%) | 4 (33%) | 4 (25%) | 2 (17%) | 0 | 1 (9%) |
| Christian Evangelical | 1 (7%) | 1 (8%) | 2 (13%) | 5 (42%) | 1 (25%) | 1 (9%) |
| Christian (other) | 2 (13%) | 0 | 0 | 0 | 0 | 0 |
| Islam | 1 (7%) | 2 (17%) | 2 (13%) | 4 (33%) | 0 | 2 (18%) |
| Residence | | | | | | |
| Urban | 9 (60%) | 11 (92%) | 10 (83%) | 12 (100%) | 4 (100%) | 9 (82%) |
| Rural | 6 (40%) | 1 (8%) | 2 (17%) | 0 (0%) | 0 (0%) | 2 (18%) |

*Other tribal ethnicities represented in our groups include Luguru, Kilindi, Kurya, Mziguwa, Mnyisanzu, Rangi, Jita, Nyambo, and Kaguru
†Includes housewives and students
‡Professional includes any salaried position (eg, nurse, teacher, government employee, etc) and retired persons
FGD, focus group discussion.

years old. Comparatively, prevalence was 30.9% (95% CI 24.7% to 37.9%) in the 40–59 age group and 11.6% (95% CI 7.6% to 17.4%) in the under 40 age group. The median BMI of individuals with hypertension was 27.4 kg/m$^2$ (IQR 24–30). Participants with hypertension were more likely to be men, older, report ongoing alcohol use, live in an urban environment, have less education and be employed as professionals ($p < 0.05$

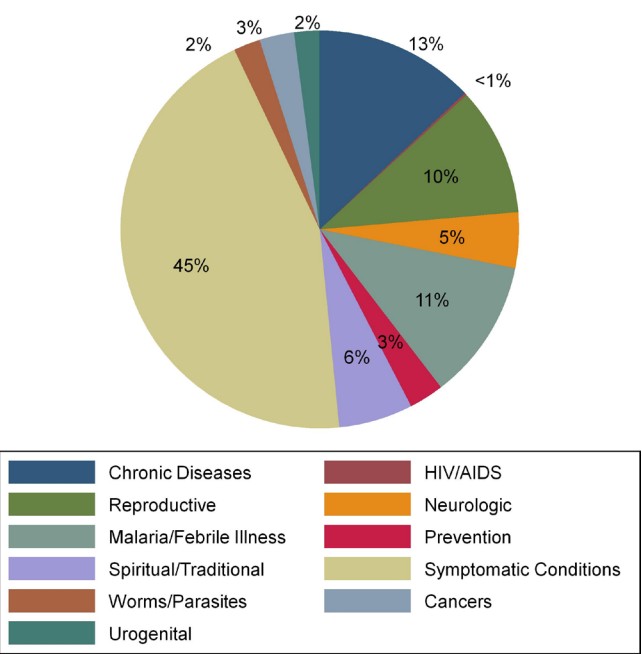

**Figure 1** Reported reasons for using traditional medicines among participants with hypertension.

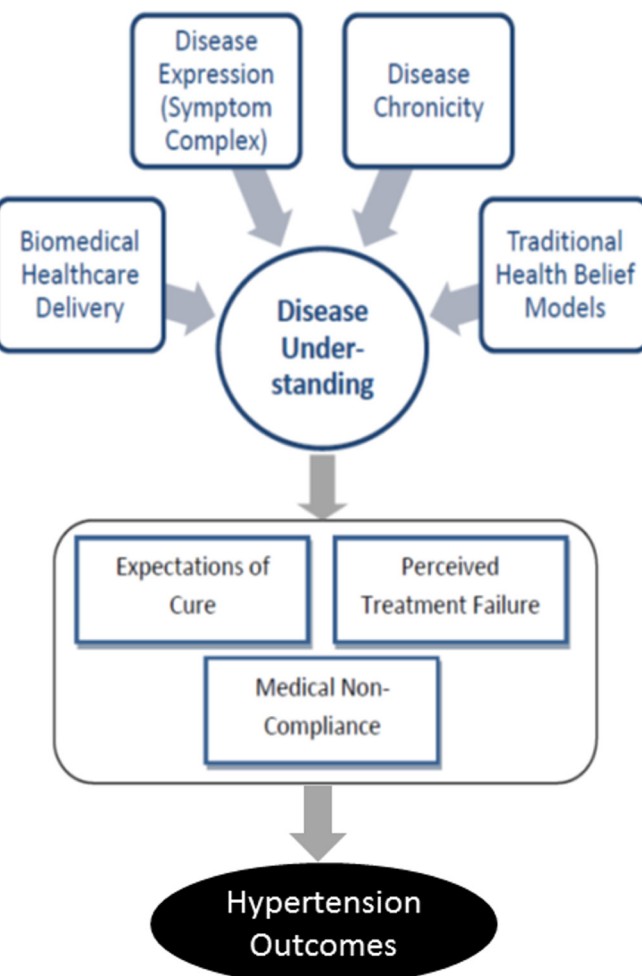

**Figure 2** Conceptual model describing the hypothesised relationship between disease understanding and hypertension outcomes.

for all) (table 1). The proportion of reported traditional medicine use among participants with hypertension was 39.3% (95% CI 30.3% to 49.1%), and the most common reasons reported for using traditional medicines were to treat daily symptomatic ailments (45%) and for treatment of chronic diseases (10%), including hypertension (figure 1).

Crude and adjusted PRRs for the relation between lifestyle-related factors and hypertension are reported in table 3. In crude models, traditional medicine use was inversely associated with hypertension prevalence (PRR 0.60; 95% CI 0.41 to 0.87), and alcohol use was significantly associated with higher prevalence of hypertension (PRR 2.29; 95% CI 1.26 to 4.15). These associations remained significant even after adjustment for age,

gender and ethnicity, with a PRR 0.37 (95% CI 0.26 to 0.54) and PRR 1.72 (95% CI 1.15 to 2.58) for traditional medicine use and alcohol use, respectively. We did not observe an association between hypertension prevalence and obesity, urban residence or tobacco use (P>0.05 for all).

### Barriers to optimal care

Despite the high disease burden, only half of participants with elevated blood pressure (48%) were aware of having hypertension. Few (23%) reported taking biomedicines for hypertension, and 12% reported taking both biomedicines and traditional medicines. Almost all participants (95%) had uncontrolled hypertension. A major theme that emerged as an important barrier for awareness and disease self-management was a difference in individual chronic disease understanding. Most notably, we identified quality or perceived quality of the biomedical healthcare delivery, disease expression, chronicity of disease and traditional health belief models as important contributors to the observed differences in chronic disease understanding, which itself contributed to unrealistic

**Table 3** Associations between lifestyle factors and hypertension; Comprehensive Kidney Disease Assessment for Risk Factors, Epidemiology, Knowledge and Attitudes, 2015

| Variables | Prevalence risk ratios (95% CI) | |
| --- | --- | --- |
| | Unadjusted | Adjusted* |
| Ongoing tobacco use | 1.25 (0.75 to 2.10) | 0.68 (0.41 to 1.14) |
| Traditional medicine use | **0.60 (0.41 to 0.87)** | **0.37 (0.26 to 0.54)** |
| Ongoing alcohol use | **2.29 (1.26 to 4.15)** | **1.72 (1.15 to 2.58)** |
| Overweight/obese | 1.00 (0.57 to 1.74) | 1.28 (0.84 to 1.97) |
| Urban residence | 0.58 (0.34 to 1.00) | 0.85 (0.59 to 1.22) |

Bold data indicates significance at the 5% level.
*Adjusted for age, gender and ethnicity.

expectations of cure, perceived treatment failures and medical non-compliance (figure 2).

Participants related structural issues and point-of-care communication issues as key barriers to optimal biomedical healthcare delivery. Structural issues in the delivery of biomedical healthcare, including long wait times, understaffing, lack of experience by healthcare providers and medication costs, were particularly concerning, and together with poor point-of-care communication between patient and provider, appeared to substantially contribute to differences in chronic disease understanding through unrealistic expectations of cure expressed as concern over ineffective or inappropriate therapies:

> MDs have a lot of patients to take care of so they lack enough time to give explanations. They hurry so that they can save as many patients as possible.

> I attended the hospital and treatment was begun but with no success for a long time; instead other parts in my body began to swell. At that time I decided to discharge myself from the hospital, and my grandmother gave me local herbals which cured me.

For many participants, differences in disease understanding were also closely related to the disease expression or the symptom complex of disease. For chronic, generally asymptomatic diseases such as hypertension, this led to unrealistic expectations of cure, greater perceptions of treatment failure by biomedicines and increased medical non-compliance:

> You know you have a disease because the body always has symptoms.

> You know you are healed as you do not have to attend the hospital anymore because your symptoms have disappeared.

For chronic diseases such as hypertension, chronicity or duration of disease was an especially salient topic closely related to differences in disease understanding. Chronic diseases were understood to be diseases that have either been untreated or undertreated, and even infectious diseases were viewed as chronic when left untreated:

> Anything that stays in the human body for a long time without being cured, like amoebae and bilharzias [schistosomiasis] is a chronic disease.

From the biomedical perspective, this different understanding of chronicity led to many challenges in achieving optimal care for patients with hypertension, particularly with respect to medication compliance, expectations of cure and perceived treatment failure. As biomedical doctors explained:

> Some patients with hypertension, after their blood pressure is controlled, they then believe they are cured after 2 or 3 months. They go to follow-up and see that their blood pressure normal; therefore, they assume they are cured and stop their medications.

The chronicity of the disease they do not understand well.

> … the people keep on seeking a 'cure' for something that is a chronic disease.

Finally, traditional health belief models were also closely associated with differences in disease understanding. Even for chronic diseases such as hypertension, participants expressed the importance of traditional medicines, and community elders and family members were considered important sources of healthcare knowledge.

> My family and I prefer not to go to hospitals. My grandparents taught us a lot (especially about plant roots) about healing and curing… my father still will not use any hospital medicines.

> … most of the chronic diseases are cured by traditional medicines.

## DISCUSSION

In a community-based setting in Northern Tanzania, we found an alarmingly high burden of hypertension. The prevalence observed was 10%–20% higher than similar regions in SSA and comparable with several LMICs, including South Africa, Brazil and China.[2 7 9 12 23–25] Despite the high burden of hypertension, awareness was low, and few had achieved optimal blood pressure control, which may be explained by observed gaps in communication, quality of healthcare delivery and traditional medicine health belief models. In particular, we identified differences in disease understanding as they relate to a disease expression through symptom complexes, disease chronicity and traditional health beliefs as potentially important barriers for achieving optimal hypertension care.

As epidemiological transitions reshape the region, Tanzania is at great risk for an explosive growth in the burden of NCDs, including hypertension.[7 26 27] The rapid pace of urbanisation and economic growth is accelerating the rate of this transition; thus, as evidenced by the high prevalence we are already observing, there is an urgent need for action.[24] Aggressive efforts should be made to diagnose and capture hypertensive patients at every single interaction within health systems. Considering the low rates of awareness in our study, all settings for diagnosis and delivery of healthcare should be explored including community centres and traditional medicine providers. Community-centred care models may be beneficial to reduce risk factors, improve treatment adherence and have been successful previously in SSA.[6 28] Given the important role of traditional medicine practices and prior willingness to refer patients to biomedical facilities for diagnostic testing,[29] partnership with traditional medicine providers should be considered to assist with risk factor reduction and care coordination, including early referral.[29 30]

Additionally, emergency department (ED)-based screening has also been successful at capturing undiagnosed/uncontrolled hypertension cases and linking patients to care in high-income countries; yet, ED-based care

models for hypertension have not been widely explored in SSA.[31–33] For example, ED-based interventions to screen for hypertension and modifiable risk factors as the first step of a care pathway may prove to be highly effective in hypertension care. Beyond diagnosis, careful consideration of the local environment and barriers to care will be necessary to create successful educational programmes and sustain hypertension control. Educational interventions should focus on the concepts of chronicity and disease expression and incorporate traditional health beliefs. Targeted and culturally tailored engagement with patients may prevent poor hypertension outcomes through improved self-management, particularly with medication compliance, expectations of cure and perceived treatment success.[34]

The application of locally tailored programmes will also require a comprehensive understanding of traditional medicine practices, particularly as they intersect with biomedical concepts of health. We found that traditional medicine use was substantially higher but associated with a lower prevalence of hypertension, and more work is needed to understand whether this could indicate a protective effect or, alternatively, a form of selection bias, reporting bias or unmeasured confounding. We also identified high concurrent use of traditional medicines and biomedicines, which further stress the importance of culturally competent interventions when addressing hypertension. Previous research by our group in the region found that traditional medicines are used by 50%–70% of the general population, consistent with other reports across Tanzania.[20 35 36] Specifically, in the current study, among those with hypertension reporting the ongoing use of biomedicine, over 60% of participants were concurrently using traditional medicines, spanning all incomes, education levels and residential settings. Although we previously identified 168 separate plant-based traditional medicines used in this region, including two used specifically for treatment of hypertension (lemon grass (*Cymbopogon citrullus*) and erabel (scientific name unknown)), further research is greatly needed to classify traditional plant-based-medicines, including their mechanisms of action, side effect profiles and potential protective effects.

In contrast to traditional medicines, we found alcohol to be positively associated with hypertension in this population, consistent with trends throughout SSA.[37–39] This region of Tanzania has a particularly high prevalence of alcohol use due to widespread cultural acceptability and home-brewing culture.[40] Given the high burden of alcohol, practitioners in all locations should be cognisant of this risk factor during both hypertension screening and treatment. Conversely, hypertension prevalence was not associated with obesity, urban residence or tobacco use, all well-established risk factors in high-income countries.[38 41] While prevalence was high in both rural and urban settings, it is unclear what factors may contribute to the chronic disease burdens across SSA, and the drivers of hypertension may be different in urban versus rural settings (eg, dietary changes, environmental exposures or differential access to care).[42 43] As with other chronic diseases, the link between obesity and

hypertension is also not well understood in SSA,[44] and it is currently unclear to what extent genetic, lifestyle and environmental factors interact to drive hypertension disparities.[26 45] As such, future work investigating the determinants and risk factors for hypertension in this setting is urgently needed.

Our study was unique in exploring hypertension in the Kilimanjaro area by using a rigorous randomly sampled, household-level survey as part of a mixed-methods design that also included qualitative sessions with key informants. We explored latent themes and social/community context for treatment failure, and by leveraging this thematic analysis, we were able to identify targetable barriers to optimal hypertension diagnosis, treatment and control. Nonetheless, we noted potential limitations to our study. First, selection bias from non-response may be present. To address any non-response bias that may have arisen from differences between the respondents and non-respondents, we used sample-balanced weights for age and gender and explored differences in occupation and education level between the two groups. In regard to internal validity, we only measured blood pressure at one setting; however, two measurements separated by >5 min were performed, and previous studies have shown that sustained elevated blood pressure in one setting may be sensitive to establish a diagnosis of hypertension.[32] Misclassification of disease around the cut-off points for hypertension may also be present although we expect this misclassification to be non-differential. Medical history was also determined by self-report for several conditions, which may be less accurate in this setting with low awareness. Also, alcohol use and tobacco use were not quantified, and future research would be strengthened by quantifying tobacco and alcohol use with validated instruments.[46] Additionally, as this was a cross-sectional study, causal inferences cannot be drawn, and associations may be influenced by confounding from unmeasured variables. Our role as biomedical practitioners may have limited our ability to interpret results (researcher bias) about differences in disease understanding, and our inferences are derived from a biomedical perspective. Finally, although we used insider–outsider coding, local non-medical surveyors, and local moderators for qualitative data collection, reporting bias may still be present.

In conclusion, in a community-based study of adults from the Kilimanjaro region of Northern Tanzania, we observed a high prevalence of hypertension, most of which was uncontrolled. Alcohol use may be an important risk factor for hypertension, and we identified several emerging cultural and social themes as barriers to optimal hypertension care, including most notably difference in disease understanding related to quality of healthcare delivery, chronicity of disease, disease expression and traditional belief models. Hypertension care models will need to leverage all existing resources from the ED, to community centres, to traditional healers in order to address the growing burden of hypertension in the region and future studies are needed to develop targeted, culturally tailored interventions designed to improve hypertension disease understanding.

**Author affiliations**
[1]Division of Emergency Medicine, Department of Surgery, Duke University, Durham, North Carolina, USA
[2]Duke Global Health Institute, Durham, North Carolina, USA
[3]Division of Global Neurosurgery and Neuroscience, Department of Neurosurgery, Duke Global Health Institute, Durham, North Carolina, USA
[4]Kilimanjaro Christian Medical University College, Moshi, Tanzania
[5]Department of Medicine, Kilimanjaro Christian Medical Center, Moshi, Tanzania
[6]Division of General Internal Medicine, Department of Medicine, Duke University, Durham, North Carolina, USA
[7]Department of Medicine, Duke Clinical Research Institute , Duke University, Durham, North Carolina, USA
[8]Division of Nephrology, Department of Medicine, Duke University, Durham, North Carolina, USA

**Acknowledgements**  We would like to thank Professors G. Ralph Corey and John Bartlett and all the staff of the KCMC-DukeCollaboration in Moshi, Tanzania for all of their efforts on the Comprehensive Kidney Disease Assessment for Risk Factors, Epidemiology, Knowledge and Attitudes (CKD-AFRiKA) study. We give a special thanks to Carol Sangawe, Cynthia Asiyo, Nicola West and Edith Aloyce Macha for their integral role in implementing the study and Jeffrey Hawley and Audrey Brown at the Duke Office of Clinical Research for their help in data management. We are also grateful to Estomih Mduma and his team for their help with Swahili translation needs.

**Contributors**  JS and UDP developed the concept of the project. JS and FK conducted the data collection. SWG and JS contributed to the writing of the manuscript. JS and JL were responsible for the analysis plan and data analysis. All authors were responsible for the final editing and approved the final manuscript.

**Funding**  This study was supported by an NIH Research Training Grant (no R25 TW009337) funded by the Fogarty International Center and the National Institute of Mental Health and a Research and Prevention Grant funded by the International Society of Nephrology Global Outreach Committee. CS would like to acknowledge salary support funding from the Fogarty International Center (K01 TW010000-01A1).

**Competing interests**  All authors have completed the ICMJE uniform disclosure form at www.icmje.org/coi_disclosure.pdf and declare: no support from any organization for the submitted work; no financial relationships with any organizations that might have an interest in the submitted work in the previous three years; no other relationships or activities that could appear to have influenced the submitted work.

**Ethics approval**  The study protocol was approved by the Duke University Institutional Review Board (no Pro00040784), the Kilimanjaro Christian Medical College (KCMC) Ethics Committee (EC no 502) and the National Institute for Medical Research (NIMR) in Tanzania.

**Provenance and peer review**  Not commissioned; externally peer reviewed.

**Data sharing statement**  We have concerns about the ethics of openly releasing the entire dataset to the public as the structure of the dataset would result in loss of participant anonymity. However, we will ensure that the dataset is openly available to researchers who contact us and meet confidentiality requirements (documentation of ethics training in conduct of human-subject research). They may contact Dr John W Stanifer, DCRI 2400 Pratt St, Durham NC 27710 or john.stanifer@duke.edu

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
