## [Reviewer comments · BMJ Open]

ARTICLE DETAILS

TITLE (PROVISIONAL)	EPIDEMIOLOGY OF HYPERTENSION IN NORTHERN TANZANIA: A COMMUNITY-BASED MIXED-METHODS STUDY
AUTHORS	Galson, Sophie; Staton, Catherine; Karia, Frank; Kilonzo, Kajiru; Lunyera, Joseph; Patel, Uptal; Hertz, Julian; Stanifer, John

VERSION 1 – REVIEW

REVIEWER	Dr Anna L Beale PhD candidate, Monash University, Melbourne Australia
REVIEW RETURNED	01-Aug-2017

GENERAL COMMENTS	Methodologically sound, interesting and well written paper. The significant negative association between traditional medicine use is interesting and could be explored further particularly in the discussion - what may be the form of selection bias in this case? What traditional medicines are used in Tanzania? This could be included in the abstract as it seems to be relevant.
--

REVIEWER	Professor Richard Walker Northumbria Healthcare NHS Foundation Trust, UK
REVIEW RETURNED	17-Aug-2017

GENERAL COMMENTS	This is an interesting and important study about the epidemiology, and understanding, of hypertension as part of a collaboration between Duke University and KCMC. It is increasingly recognised that non-communicable diseases (NCDs) are a major, and increasing burden, in low and middle income countries (LMIC) such as Tanzania and hypertension is a very important risk factor which is currently under diagnosed and, even when diagnosed, sub-optimally treated for various reasons. The quantitative component of the research was based on a cross-sectional household epidemiological survey as part of the CKD-AFRiKA Study and the random selection of the households in Moshi urban and Moshi rural districts is well described. The qualitative component of the study was based on 5 focus group discussions and 11 in-depth interviews. My comments on the paper are below: Abstract In the Methods section it says the study was from January 2014 to January 2015 and yet in the main paper it says December 2013 to June 2015. This inconsistency needs to be clarified.
---

Methods

The quantitative sampling methods are well described. The investigators made a minimum of 2 additional visits if they didn't get permission at the first visit and also made multiple phone calls.

Demographic data were collected. Also, a medical history survey was conducted which included self-reported history of different medical conditions and any regular medication. It may be worth reporting in the Limitations sections that self-reported history may be less reliable in LMICs where often patients do not have access to investigation and diagnosis for medical conditions as readily as in other areas of the world.

In relation to measurement of hypertension it is stated that blood pressure was measured after 5 minutes of participants sitting. In the Discussion it is stated that a further blood pressure measurement was taken 5 minutes later and this should be mentioned in the Methods section.

The qualitative data collection and analysis seems entirely appropriate and robust. The investigators used 2 note-takers to transcribe and independently translate each session with a moderator then reviewing the accuracy of the transcripts. The coding was then carried out by a cultural insider and a cultural outsider to allow exploration of concepts that otherwise may have been overlooked or misinterpreted by either researcher individually. I would commend the authors on this.

Since alcohol use was significantly related to hypertension I think it would be helpful to know whether any effort was made to quantify the alcohol use as there is potentially a big difference between heavy users and occasional users?

The authors found that traditional medicine use was associated with lower prevalence of hypertension. Do they have any more details on frequency and type of traditional medicine, and also if this was actually for hypertension or for something else?

Results

Non-responders were more likely to be men and young adults such that women actually represented 74% of the 481 participants. There was a wide age range and hypertension was, not surprisingly, more common in those aged over 60. It is well known that hypertension prevalence rates increase with age but it would be interesting to know if some of the younger age groups are showing higher rates of hypertension. However, as the confidence intervals are quite wide for the whole group I suspect that sub-group analysis may be quite difficult due to the numbers available.

It is quite surprising that hypertension prevalence was not associated with obesity, urban residence or tobacco. Have the authors got any suggestions as to why this might be?

The section about barriers to optimal care based on the qualitative research is very interesting and supports previous research about the difficulties of providing long term treatment for hypertension for which there is no "cure". I thought some of the quotes were very illuminating.

	Discussion Contact with traditional healers was associated with lower prevalence of hypertension and the authors suggest that traditional healers may help with early identification and referral. Have they discussed with traditional healers whether they would be interested to do this? Would they be prepared to measure blood pressure? Would they be able to participate in follow up? The authors have discussed most of the potential limitations of the study. References Reference 4 and reference 34 do not appear to have years.
--	---

REVIEWER	Kelly D. Taylor, PhD, MPH, Research Scientist University of California San Francisco United States
REVIEW RETURNED	22-Aug-2017

GENERAL COMMENTS	This is an interesting mixed methods study to investigate the epidemiology of hypertension and barriers to care among rural and urban adults in Northern Tanzania. The manuscript was well written and the study appropriately designed to address the research objectives. The authors offer actionable recommendations to address the high burden of hypertension in this area. Specific comments 1. Methods  • In the Ethics Statement it would be informative if the authors include the amount of the participant reimbursement in local currency and the US dollar equivalent at the time of the study. • Was any reimbursement provided to the qualitative study participants? 2. Study setting  • State why the urban and rural Moshi districts were selected for this study out of the seven districts in the region. 3. Qualitative data collection  • If the focus group discussions were conducted in both urban and rural settings, state the urban/rural breakdown of the participants. If not, include the rationale for conducting the qualitative interviews in a single setting. • How were the participants recruited?
---

VERSION 1 – AUTHOR RESPONSE

Reviewer: 1

Reviewer Name: Dr Anna L Beale

Institution and Country: PhD candidate, Monash University, Melbourne Australia

Comment: The significant negative association between traditional medicine use is interesting and could be explored further particularly in the discussion - what may be the form of selection bias in this case? What traditional medicines are used in Tanzania? This could be included in the abstract as it seems to be relevant.

Response: Thank you. We also have been very interested in the role and cultural importance of traditional medicine use, particularly around chronic conditions such as hypertension. As part of the CKD-AFRIKA study, we have investigated several aspects of traditional medicine use, and we have expanded the Discussion section (lines 349-362) to highlight some of these important topics. We have also included text in the abstract on lines 66-67.

Reviewer: 2

Reviewer Name: Professor Richard Walker

Institution and Country: Northumbria Healthcare NHS Foundation Trust, UK

Comment: Abstract

In the Methods section it says the study was from January 2014 to January 2015 and yet in the main paper it says December 2013 to June 2015. This inconsistency needs to be clarified.

Response: Thank you. We apologize for the error, which has been corrected. The methods section of the abstract was corrected to December 2013 (line 54).

Comment:Methods

It may be worth reporting in the Limitations sections that self-reported history may be less reliable in LMICs where often patients do not have access to investigation and diagnosis for medical conditions as readily as in other areas of the world.

Response: This important point has been added to the Discussion section (lines 388-390).

Comment: In relation to measurement of hypertension it is stated that blood pressure was measured after 5 minutes of participants sitting. In the Discussion it is stated that a further blood pressure measurement was taken 5 minutes later and this should be mentioned in the Methods section.

Response: Thank you. This detail was added to the methods section (lines 148-149).

Comment: Since alcohol use was significantly related to hypertension I think it would be helpful to know whether any effort was made to quantify the alcohol use as there is potentially a big difference between heavy users and occasional users?

Response: Thank you. Alcohol use was defined as self-reported current and ongoing use of alcohol products, former use of alcohol, or no use ever of alcohol. This has been added to the Methods section (lines 152-153). Unfortunately, we did not quantify use of alcohol at the time of the initial survey. We have added text in the Discussion (lines 390-391) highlighting this important limitation and the need for further investigations into this association.

Comment: The authors found that traditional medicine use was associated with lower prevalence of hypertension. Do they have any more details on frequency and type of traditional medicine, and also if this was actually for hypertension or for something else?

Response: Yes, additional details are available and have been added to the Results and Discussion sections (lines 215-218, 240-243, and 349-362).

Previous research conducted by this group has demonstrated that 50-70% of the population uses traditional medicines (TM) and the most common reasons for TM use are chronic diseases (such as hypertension) and symptomatic ailments (Stanifer JW and Lunyera J, et al. Traditional Medicine Practices among Community Members with Chronic Kidney Disease in Northern Tanzania: An Ethnomedical Survey. BMC Nephrology 2015; 16:170). The frequency of use among most adults with chronic diseases (including hypertension) was 1–5 times (31.0 %; 95 % CI 20.8–43.3 %) or 6–10 times (11.5 %; 95 % CI 5.50–22.4 %) per year. The prevalence of TM use of more than ten times per year was 7.10 % (95 % CI 3.60–13.5 %). Among those with hypertension, concurrent use of TM and biomedicines was substantial, with a prevalence of 8.4% (3.2-20.2%), and among those with hypertension reporting the ongoing use of biomedicine for any condition (n=32), the prevalence of TM use was 76% (47.6-91.7%). Among those reporting the ongoing use of biomedicine specifically for treatment of hypertension (n=23), the prevalence of TM use was 81.5% (51.6-94.8%). However, among those without hypertension, concurrent use of TM and biomedicine was 5.6% (3.4-9.3%).

Comment: Results

Non-responders were more likely to be men and young adults such that women actually represented 74% of the 481 participants. There was a wide age range and hypertension was, not surprisingly, more common in those aged over 60. It is well known that hypertension prevalence rates increase with age but it would be interesting to know if some of the younger age groups are showing higher rates of hypertension. However, as the confidence intervals are quite wide for the whole group I suspect that sub-group analysis may be quite difficult due to the numbers available.

Response: Yes, hypertension was most prevalent in those 60 years old and above (59.3% (95% CI 50.1-67.9%) but also common in younger and middle age groups as well. The prevalence in the under 40 age group was 11.6% (95% CI 7.6-17.4) and in the 40-59 age group was 30.9% (95% CI 24.7-37.9). These details have been reported in the manuscript (lines 236-238).

Comment: It is quite surprising that hypertension prevalence was not associated with obesity, urban residence or tobacco. Have the authors got any suggestions as to why this might be?

Response: Thank you. Prevalence was high in both rural and urban settings, 33.3% (25.1-42.7) and 30.3% (25.8-35.2%) respectively. It is unclear what factors may contribute to the chronic disease burdens across sub-Saharan Africa, and the drivers of hypertension may well be different in urban vs. rural settings (e.g. dietary changes, environmental exposures, or differential access to care). Studies investigating novel mechanisms are urgently needed. As with other chronic diseases (e.g. CKD), the link between obesity and hypertension is not well understood in sub-Saharan Africa, and it is unclear to what extent or how genetic, lifestyle, and environmental factors interact to drive hypertension disparities. We have expanded the discussion (lines 368-376) to include these important considerations.

Comment: Discussion

Contact with traditional healers was associated with lower prevalence of hypertension and the authors suggest that traditional healers may help with early identification and referral. Have they discussed with traditional healers whether they would be interested to do this? Would they be prepared to measure blood pressure? Would they be able to participate in follow up?

Response: Thank you, we have added details to the discussion to clarify this point. Our goal was to highlight potential avenues of future research and collaboration (lines 336-337).

Comment:References

Reference 4 and reference 34 do not appear to have years.

Response: Thank you, the year for reference 4 was added and spacing was corrected for reference 34.

Reviewer: 3

Reviewer Name: Kelly D. Taylor, PhD, MPH, Research Scientist

Institution and Country: University of California San Francisco, United States

1. Methods

Comment: In the Ethics Statement it would be informative if the authors include the amount of the participant reimbursement in local currency and the US dollar equivalent at the time of the study.

Response: Thank you, we reimbursed participants between 1500-12,000 TSH (~\$0.75-\$5.00 USD) depending upon their distance of travel. We have added this information on lines 114-115.

Comment: Was any reimbursement provided to the qualitative study participants?

Response: Thank you, reimbursement was the same for qualitative and quantitative participants. We reimbursed participants between 1500-12,000 TSH (~\$0.75-\$5.00 USD) depending upon their distance of travel. We have added this information on lines 114-115.

2. Study setting

Comment: State why the urban and rural Moshi districts were selected for this study out of the seven districts in the region.

Response: The Moshi Urban and Rural districts were selected based on their representative population, similar to the surrounding region, as well as their proximity to the central laboratory and our research infrastructure. We have made these points more clear in the text on lines 123-124.

3. Qualitative data collection

Comment: If the focus group discussions were conducted in both urban and rural settings, state the urban/rural breakdown of the participants. If not, include the rationale for conducting the qualitative interviews in a single setting.

Response: The focus group were physically held in an urban setting, but we enrolled participants living in both urban and rural communities. The breakdown has been added to table 2 and lines 229-230.

Comment: How were the participants recruited?

Response: Thank you. For the qualitative focus groups, purposive sampling was used to recruit the key informants.

We have provided more details in the text (lines 161-166).

VERSION 2 – REVIEW

REVIEWER	Anna L Beale Monash University, Australia
REVIEW RETURNED	18-Sep-2017

GENERAL COMMENTS	Thank you for this revision. Your discussion of the association between traditional medicine use and lower rates of hypertension is now much more complete. My only remaining comment is that reference 3 has an incomplete link, which should be rectified prior to publication.
--

REVIEWER	Professor Richard Walker Northumbria healthcare NHS Foundation Trust, UK
REVIEW RETURNED	18-Sep-2017

GENERAL COMMENTS	The authors have addressed all my concerns satisfactorily. NB typos line 216 Among participants... line 351 ...substantially higher... line 374 ...environmental factors...
--

REVIEWER	Kelly D Taylor, PhD, MPH, Research Scientist University of California, San Francisco, USA
REVIEW RETURNED	29-Sep-2017

GENERAL COMMENTS	The authors have adequately addressed the comments/recommendations to: 1) Include the amount of the participant reimbursement 2) State the rationale for selecting the study setting 3) Include the urban/rural breakdown of the study participants 4) State the method of participant recruitment
--

VERSION 2 – AUTHOR RESPONSE

Reviewer: 1

Reviewer Name: Dr Anna L Beale

Institution and Country: PhD candidate, Monash University, Melbourne Australia

Comment 1. My only remaining comment is that reference 3 has an incomplete link, which should be rectified prior to publication.

Response: Thank you, the link for reference 3 was corrected (line 446)

Reviewer: 2

Reviewer Name: Professor Richard Walker

Institution and Country: Northumbria Healthcare NHS Foundation Trust, UK

Comment 1. line 216 Among participants...

Response: Participants spelling error was corrected (line 216).

Comment 2. line 351 ...substantially higher...

Response: Substantially high was changed to substantially higher (line 354).

Comment 3. line 374 ...environmental factors...

Response: Environmental was changed to environmental factors (line 375).

Reviewer: 3

Reviewer Name: Kelly D. Taylor, PhD, MPH, Research Scientist

Institution and Country: University of California San Francisco, United States

Comment 1. No further suggested revisions.

Response: Thank you.